# How Diabetes and Other Comorbidities of Elderly Patients and Their Treatment Influence Levels of Glycation Products

**DOI:** 10.3390/ijerph19127524

**Published:** 2022-06-20

**Authors:** Aleksandra Kuzan, Emilia Królewicz, Irena Kustrzeba-Wójcicka, Karolina Lindner-Pawłowicz, Małgorzata Sobieszczańska

**Affiliations:** 1Department of Medical Biochemistry, Wroclaw Medical University, 50-368 Wroclaw, Poland; emilia.krolewicz@umw.edu.pl (E.K.); irena.kustrzeba-wojcicka@umw.edu.pl (I.K.-W.); 2Clinical Department of Geriatrics, Wroclaw Medical University, 50-369 Wroclaw, Poland; karolina.lindner@umw.edu.pl (K.L.-P.); malgorzata.sobieszczanska@umw.edu.pl (M.S.)

**Keywords:** aging, geriatric care, elderly diseases, diabetes mellitus type 2, glycation markers

## Abstract

Medical care for geriatric patients is a great challenge, mainly due to various overlapping deficits relevant to numerous coexisting diseases, of which the most common are diabetes mellitus and atherosclerosis. In the case of diabetes, the glycation process is intensified, which accelerates atherosclerosis development and diabetic complications. Our goal was to investigate the relationship between the classical biochemical parameters of diabetes and atherosclerosis, as well as parameters which may indicate a nephropathy, and the parameters strictly related to glycation, taking into account the pharmacological treatment of patients. Methods: We analyzed the patients’ serum concentrations of fluorescent glycation product—pentosidine, concentrations of soluble receptors for advanced glycation products (sRAGE), lipoprotein receptor-1 (LOX-1), galectin 3 (GAL3), scavenger receptor class A (SR-A), and scavenger receptor class B (SR-BI), as well as the level of lipid peroxidation and free amine content. Among the identified correlations, the most interesting are the following: sRAGE with triglycerides (r = 0.47, *p* = 0.009), sRAGE with SR-BI (r = 0.47, *p* = 0.013), SR-BI with LOX-1 (r = 0.31, *p* = 0.013), and SR-BI with HDL (r = −0.30, *p* = 0.02). It has been shown that pentosidine and reactive free amine contents are significantly higher in elderly patients with ischemic heart disease. Pentosidine is also significantly higher in patients with arterial hypertension. Malondialdehyde turned out to be higher in patients with diabetes mellitus type 2 that was not treated with insulin or metformin than in those treated with both medications (*p* = 0.052). GAL3 was found to be lower both in persons without diabetes and in diabetics treated with metformin (*p* = 0.005). LOX-1 was higher in diabetic patients not treated with metformin or insulin, and lowest in diabetics treated with both insulin and metformin, with the effect of metformin reducing LOX-1 levels (*p* = 0.039). Our results were the basis for a discussion about the diagnostic value in the clinical practice of LOX-1 and GAL3 in geriatric patients with diabetes and also provide grounds for inferring the therapeutic benefits of insulin and metformin treatment.

## 1. Introduction

In 1909, the Viennese doctor Ignatz Leo Nascher introduced the term “geriatrics”, which meant “old age medicine” [1]. As a pioneer of this clinical discipline, he justified the need to separate geriatrics from adult medicine [2]. In 2004, the World Health Organization (WHO) defined geriatrics as a field of medicine dealing with diseases and healthcare for the elderly [3].

The number and proportion of people aged 60 years and older in the global population is increasing. In 2019, the number of elderly people was 1 billion, and this number will increase to 1.4 billion by 2030 and 2.1 billion by 2050 [4]. This increase is occurring at an unprecedented pace and will accelerate in the coming decades, particularly in developing countries. The explanation for this phenomenon is mainly the improvement of life expectancies, the high quality of health care, the post-World War 2 baby boom [4], and declining total fertility rates [5]. Seniors suffer from functional and mental decline, cognitive deficits, frailty, and multiple comorbidities, resulting, as a consequence, in polypharmacy. This is why it is so important to develop geriatrics as an interdisciplinary and multidimensional field of medicine using a comprehensive approach for elderly patients.

Clinical manifestations of diseases affecting elderly persons include a lot of somatic and mental problems. This applies to atherosclerotic cardiovascular disease, diabetes, arterial hypertension, osteoarthritis, urinary incontinence, hearing problems, and visual impairment, as well as osteoporosis and sarcopenia. Moreover, dementia, including Alzheimer’s disease and depression, cannot be omitted [6]. Furthermore, there is the association between chronic low-grade inflammation (“inflammaging”) and many adverse health outcomes in elderly patients, including chronic diseases, functional decline, and mortality [7]. 

Diabetes mellitus (DM), in particular type 2 diabetes (T2DM), is the most common disease in older adults. It is estimated that T2DM affects up to 40 percent of people over 65 years old in developed and even in developing countries [8]. Approximately 20 percent of older persons suffer from clinically apparent DM, and a similar proportion have undiagnosed DM [9]. Fueled in part by the obesity epidemic, this number is projected to double in the next 20 years and quadruple by 2050 [10]. Diabetes, especially nowadays, is a huge threat to older people because of the pandemic related to the emergence of SARS-CoV-2 and COVID-19. As a result, worse access to health care and physicians and the lack of physical activity, associated with being confined at home, are observed. These factors contribute to carbohydrate metabolism disorders and insufficient glycemic control [11,12]. 

Chronically sustained high blood sugar levels lead to the damage and abnormal function of many organs, especially, the heart, blood vessels, kidneys, nerves, and eyesight [13,14]. Diabetes mellitus is associated with an increased level of free radicals, disturbances of the enzymatic antioxidant defense system, and lower a concentration of exogenous antioxidants. In consequence, these abnormalities lead to a redox imbalance called oxidative stress [15]. 

In diabetes mellitus, the theory of oxidative stress is associated with autooxidation of glucose (glycoxidation), which produces reactive ketoaldehydes. This intensifies the process of non-enzymatic protein glycation leading to formation of advanced glycation end products (AGEs). During this process, due to the strong dependence on oxygen, many toxic oxygen derivatives are formed. In people with diabetes, the serum concentrations of pentosidine, pyraline, or carboxymethyl lysine are increased. The severity of the production of reactive oxygen species (ROS) depends on the degree of protein glycation and thus on glycemia. Many proteins in the body undergo the process of glycation, including proteins of the basement membrane and blood proteins [16].

In the present study, a focus was placed not only on AGEs but also on their receptors. Glycation products are ligands for numerous proteins, the most important of which appear to be: receptor for advanced glycation end products (RAGE; synonym: AGER), scavenger receptors class A (SR-A; synonyms: MSR1, SCARA1, macrophage acetylated LDL receptor I and II, scavenger receptor class A member 1) and class B, type I (SR-BI; synonyms: SCARB1, CD36L1; SRB1; CLA1; collagen type I receptor thrombospondin receptor-like 1; CD36 and LIMPII analogous 1), lectin-like oxidized low-density lipoprotein receptor 1 (LOX-1), and galectin 3 (GAL3) [17,18,19,20,21].

Gaining detailed knowledge about the factors leading to the aging process and the course of diseases occurring in the elderly (including diabetes) makes it possible to develop a way to delay the decline in the functionality of the human body and at the same time extend life. Our goal was to investigate the relationship between the classical biochemical parameters of diabetes and atherosclerosis and the parameters strictly related to glycation, taking into account the hypoglycemic treatment of patients. This work can be called a preliminary study, allowing to identify directions for further research promising to demonstrate cause−effect relationships in geriatric diseases.

## 2. Materials and Methods

The study group consisted of patients hospitalized at the Department of Geriatrics, University Teaching Hospital, in Wrocław. Included in this observational prospective study were blood samples of 79 consecutive patients, who gave conscious permission and fulfilled the inclusion criteria. The inclusion criteria for the study group were: age over 60 and the ability to express informed consent in the study. The exclusion criteria were: acute conditions in the 4 weeks prior to study participation, such as myocardial infarction, venous thromboembolism, stroke; as well as cancer. In total, 79 patients who were qualified for the scientific project gave written permission for participation in the study and taking biological materials. The Bioethical Commission at Wroclaw Medical University gave written permission (opinion number KB-344/2017). The results of routine laboratory tests, such as serum levels of: glucose, total protein, hemoglobin, glycated hemoglobin (HbA1c), C-reactive protein (CRP), cholesterol, HDL, LDL, triglycerides, creatinine, and uric acid, as well estimated glomerular filtration rate (eGFR, using MDRD formula [22]), were used for statistical analyses. Forty-one diabetic patients were assessed in four groups: (1) treated with metformin only, (2) treated with insulin only, (3) treated with both medications, and (4) treated with other hypoglycemic drugs (glimepiridum, gliclazide, linagliptin, liraglutide). Atherosclerosis was diagnosed on the basis of the disclosure of atherosclerotic changes in vascular areas, e.g., in the cerebral arteries, in imaging studies. The characteristics of the study population are presented in Table 1. Reagents, unless indicated otherwise in the text, were purchased from POCH (Gliwice, Poland) and Sigma Aldrich (Saint Louis, MO, USA).

### 2.1. Analysis of Pentosidine Content Using Fluorometric Method 

Pentosidine is a ribose-derived glyco-oxidation product of arginine and lysine residues and is one of the fluorescent forms of AGEs. The methodology of pentosidine determination was a slight modification of that of Leszek et al. [23]. Serum samples were diluted 100-fold with 0.9% NaCl. The absorbances of these samples at λ = 280 nm were measured. To assess the total pentosidine content in the sample, the fluorescence of the samples was measured at excitation wavelength 335 nm and emission 385 nm on an EnSpire Multimode Plate Reader (Perkin Elmer, Waltham, MA, USA). Fluorescence was divided by Abs 280 assuming the resulting values as data expressed in arbitrary units.

### 2.2. Assessment of Content of Compounds with Thiol Groups (Glutathione)

Determination of the glutathione (one of main factors responsible for the antioxidative mechanism in the body) concentration is usually performed as a measure of thiol groups content using Ellman’s reagent (5,5’-dithio-bis-(2-nitrobenzoic acid); DTNB). The method was developed in 1991 by Diplock [24] et al. To achieve this goal, a reaction between compounds containing thiol groups from patients’ serum and 0.1 M DTNB in 10 mM sodium-phosphate buffer pH 8.00 was conducted with the presence of 10% SDS. Parallel to this, an analogical reaction with glutathione of concentration 0.25–1 mM was performed in order to draw a standard curve. The reaction was conducted for 60 min at 37 ℃. In the next step, an absorbance against blank sample at λ = 412 nm was measured, and on the basis of data from the standard curve, the content of thiol group was calculated in the examined material.

### 2.3. Assessment of Lipid Peroxidation on the Basis of Determination of Malondialdehyde (MDA) in Serum

A method of estimation of lipid peroxidation intensity is based on the reaction of malondialdehyde with thiobarbituric acid (TBA). The pink product obtained was later determined by spectrophotometry [25]. The reactive mixtures consisted of serum, 15% TCA acid solution in 0.25 M HCl, and 0.37% TBA in 0.25 M HCl. The mixtures were incubated for 20 min at 100 °C, then cooled down and rotated for 5 min at 4700 RPM. Absorbance of supernatant was measured at 535 nm. Calculation of MDA concentration was performed using the Lambert−Beer law at absorbance coefficient ε = 156 mmol^−1^·L·cm^−1^.

### 2.4. Reactive Free Amine Content

The procedure is described in the work of Fracasso et al. [26]. Serum samples were diluted 200-fold with 50 mM carbonate buffer (pH 10.5). Then, 9 μL aliquots of the diluted specimens were pipetted into a 96-well microplate, and 91 μL of freshly made o-phthalaldehyde (OPA) solution (5 mg OPA, 100 μL pure ethanol, 5 μL β-2-mercaptoethanol, and 10 mL 50 mM carbonate buffer pH 10.5) was added in all wells. Samples were read immediately at 340 nm excitation and 455 nm emission wavelengths.

### 2.5. Analysis of GAL3 and LOX1 Protein Content

The concentration of GAL3 and LOX1 in serum samples was performed using human immunoassay ELISA kits from Elabscience^®^ (Elabscience Biotechnology Inc., Houston, TX, USA). The manufacturer’s instructions were followed.

### 2.6. Analysis of sRAGE, SCRAB1, and MSR1 Protein Content

The concentration of sRAGE, SCRAB1, and MSR1 in serum samples was performed using human immunoassay ELISA kits from ELK Biotechnology (ELK (Wuhan) Biotechnology CO., Ltd., Wuhan, China). The manufacturer’s instructions were followed.

### 2.7. Statistical Analysis

Statistical analysis was conducted using the data analysis software system Statistica (version 13.3, StatSoft, TIBCO Software Inc., Palo Alto, CA, USA). Normality of distribution and homogeneity variance of parameters were checked, and if the parameters were not normal in distribution, a nonparametric test was chosen. Frequency analysis was conducted using Fisher’s exact test or χ^2^ test. For comparison of all the variables between groups (with diabetes/without diabetes; with hypertension/without hypertension; with ischemic heart disease (IHD)/without IHD; etc.), the Mann−Whitney U test was used. A Kruskal–Wallis test with post-hoc median test was used when comparing more than two continuous variables. All calculated *p*-values were two-sided, and the statistical significance level was set at *p* < 0.05. Spearman rank correlation analysis was used to determine associations between oxidative and inflammatory markers with selected parameters. To determine the most important predictors of AGE, pentosidine, and SR-BI levels, modeling with Generalized Linear Models (GLM) was used. As covariates in the models, all those variables which showed significant relationship with the dependent variable were introduced in the univariate analysis. The variables were fitted using Gaussian distribution with the identity link function. The models were constructed in a two-way step procedure based on the Wald test results with the significance level increased to 20% and Akaike information criterion (AIC). Statistics for the variables rejected during modeling, which are given in the tables, were obtained after their independent inclusion in the final model.

## 3. Results

### 3.1. Average Values and Correlations between Variables

The studied geriatric group is heterogeneous due to various disease states that affect individual patients (Table 1). Therefore, relatively large standard deviations in the analyzed parameters are also observed (Table 2).

Appendix A (Appendix A) shows what correlations are found between the analyzed parameters. It is in the form of a heat map, where the redder the color, the stronger the positive correlation; the more blue, the stronger the negative correlation. Due to the number of parameters, not all of them are listed, but only those that seem to be the most interesting and important, i.e., have the highest r-factor at *p* < 0.005 and are relatively little or not described in the literature on the subject, are emphasized, as given below:

Galectin 3 content correlates positively with HbA1C (r = 0.267, *p* = 0.031, Figure 1A), with BMI (r = 0.62, *p* < 0.0001, Figure 1B), as well as with glucose (r = 0.39, *p* < 0.0001), with triglycerides (r = 0.36, *p* = 0.002), and negatively with GFR (r = −0.32, *p* = 0.006, Figure 1C).

The content of fluorescing pentosidine negatively correlates with GFR (r = −0.36, *p* = 0.02, Figure 1D), positively correlates with creatinine (r = 0.26, *p* = 0.24, Figure 1E), and negatively with LOX1 (r = −0.23, *p* = 0.045, Figure 1F). sRAGE content correlates with triglycerides (r = 0.47, *p* = 0.009, Figure 1G) and SR-BI (r = 0.47, *p* = 0.013). The latter additionally correlates with HDL (r = −0.30, *p* = 0.02, Figure 1H) and with LOX-1 (r = 0.32, *p* = 0.013, Figure 1I).

Glutathione and SR-A did not correlate with any parameters.

### 3.2. Relationships between Diseases and Biochemical Parameters

Pentosidine has been shown to be higher in people with ischemic heart disease (IHD) (*p* = 0.0018) (Figure 2A). Likewise, reactive free amine content is higher in people with IHD (*p* = 0.0491) (Figure 2B). It was also shown that hypertensive patients have significantly elevated levels of pentosidine (*p* = 0.003) (Figure 2C).

None of the analyzed parameters were significantly associated with the history of myocardial infarction, but one of them, namely reactive free amine content, was significantly increased in post-stroke patients (*p* = 0.009) (Figure 2D).

### 3.3. Influence of Diabetes Therapy on Parameters Related to Glycation and Peroxidation

Serum GAL3 content is found to be lower in nondiabetic subjects (*p* = 0.0047), and in the diabetic group it is lower in subjects treated with metformin as compared with diabetics treated with insulin or treated with drugs other than metformin and insulin (*p* = 0.0268) (Figure 3A,B).

LOX1 is high in diabetic patients not treated with insulin or metformin and statistically significantly lower in people treated with both insulin and metformin compared to people treated with insulin alone (*p* = 0.0398) (Figure 3C). The median for the result of diabetics treated separately with insulin and separately with metformin is lower than for those not treated with insulin or metformin; probably due to the small number of patients not treated with insulin or metformin, the statistical significance was not obtained here.

MDA is higher in diabetic patients not treated with insulin or metformin than those treated with metformin and insulin simultaneously (*p* = 0.052) (Figure 3D). The median for those treated with one drug is also lower than for those not treated, but statistical significance was also not obtained here.

### 3.4. Statistical Modeling Results

In the multivariate analysis of SR-BI levels based on HDL and uric acid, as the variables, only HDL turned out to be a significant predictor (Wald χ^2^ 6.60; df 1; *p* = 0.01).

The following variables were used in the multivariate analysis of pentosidine levels: IHD, hypertension, GFR, creatinine, and HbA1c. The best model included three predictors: IHD, hypertension, and GFR, of which the first two were barely significant (IHD Wald χ^2^ 3.911; df 1; *p* = 0.048; hypertension Wald χ^2^ 3.563; df 1; *p* = 0.59; GFR 1.644 df 1, *p* = 0.2).

## 4. Discussion

Currently, glycated hemoglobin (HbA1c) is the only marker of glycation that is routinely used in diagnostics. It is considered as the gold standard in the estimation of long-term glycemia because it is considered to reflect a weighted measure of the average blood glucose levels during the past 120 days and has been shown to correlate with adverse outcomes in diabetics, especially related to microvascular complications [27,28]. The advantages of HbA1c include less day-to-day variability and greater convenience as fasting is not required compared to fasting plasma glucose measurements and oral glucose tolerance tests. However, measured HbA1c is dependent on red blood cells’ life spans, which may vary among individuals. In the context of geriatric patients, it is important that an increase of HbA1c is observed with increasing age in nondiabetic individuals [28]. Furthermore, we want to emphasize that HbA1c is not the de facto product of advanced glycation, but rather a product of Amadori rearrangement—what can be called an early glycation product [29]. In the present study, we propose a new approach by analyzing several glycation parameters: pentosidine, as well as four different types of receptors for AGEs and two ancillary parameters related to oxidative stress and peroxidation that accompany aging and glycation.

The expected result was the observation of a correlation between pentosidine and HbA1c. This effect has not been observed. Instead, we found a correlation between parameter pairs: HbA1c-GAL3 and HbA1c-BMI. Lack of correlation between serum pentosidine and HbA1c was reported previously by, for example, Hashimoto et al. or Sato et al. [30,31]. It can be seen that although both parameters are closely related to glycation, their concentrations change independently of each other. The relationship between HbA1c and BMI is not surprising, as people with unregulated diabetes are often obese. The HbA1c-GAL3 correlation turns out to be quite an interesting result, hence the considerations of what this information contributes to the diagnosis of older people.

Galectin-3 is a protein from the group of β-galactoside binding lectins with pro-inflammatory properties, presented as a novel biomarker of pathological conditions, such as various types of cancer, especially thyroid cancer [17], acute and chronic heart failure [32,33], chronic pancreatitis [18], prediabetes state and apparent diabetes [34], depression [35], and many others. It has multiple functions including cell adhesion, inflammation, extracellular matrix formation, differentiation, proliferation, embryogenesis, and host–pathogen interactions [35,36,37]. As a lectin, it binds carbohydrates but it is characterized by a wide range of ligands and, in addition to simple sugars and disaccharides such as galactose, lactose, or N-acetyllactosamine, it also binds large carbohydrate molecules, such as cell surface N-glycans and advanced glycation products that are intensely generated during hyperglycemia [17,18]. Moreover, the protein is found to cause cellular and systemic insulin resistance [37]. Therefore, we expected that galectin-3 would also reflect the various pathological states typical for the geriatric population.

The correlation of galectin-3 with BMI values has been demonstrated repeatedly in different study groups [36,37,38]. Our study confirmed this result. Similar correlations with glucose and triglycerides have been previously described, and these relationships are especially often observed in women with polycystic ovary syndrome and pregnant women with preeclampsia [38,39]. These correlations were also described based on studies on cardiac patients [33] and with hypothyroidism [40]. We describe these relationships probably for the first time for geriatric persons.

According to our observations, galectin-3 inversely correlates with GFR, indicating a relationship with kidney damage. Many studies have already concluded that galectin-3 strongly affects this organ’s fibrosis, and it has even been suggested that its measurement will allow to predict renal failure and patients’ survival [41,42].

Moreover, galectin 3 affects fibrosis of many other organs, including the heart after myocardial infarction (MI), hence the need to lower the level of this glycoprotein in order to avoid heart failure after MI [43]. Our studies show that metformin is a drug that is effective in reducing plasma GAL3 levels. Our results are consistent with the results of Asensio-Lopez et al. and Weigert et al. [43,44]. The practical conclusion is that the administration of metformin in geriatric patients is beneficial not only in regulating carbohydrate metabolism but also in reducing the likelihood of heart failure threatening elderly persons.

Another issue is the previously proven thrombogenic effect of galectin-3 [45,46]. The prothrombotic effect of GAL3 may contribute to higher mortality in geriatric patients due to cardiovascular diseases. It is postulated that galectin-3 inhibitors can be used as novel antithrombotic drugs [46]. In addition, the aforementioned metformin has an effect, inter alia, of a lower thrombogenicity, and treatment with this drug is associated with reduced tissue factor procoagulant activity in patients with poorly controlled diabetes [47]. It is concluded that the molecular mechanism of metformin’s action in this context may be related to the effect on galectin, although a detailed study in this direction should be carried out.

In our study, galectin 3 is significantly higher in elderly patients with diabetes type 2 compared with patients without diabetes. It would seem that galectin 3 is a very interesting diagnostic tool. Many authors present it as a novel biomarker—for example, Hrynchyshyn et al. and Kanukurti et al. present it as a new biomarker for the diagnosis, analysis, and prognosis of acute and chronic heart failure [32,33]; King et al. present it as a novel inflammatory biomarker related with depression symptom severity [35]; and Alves et al. name it as a potential biomarker to insulin resistance and obesity in women with polycystic ovary syndrome [38].

However, doubts as to its diagnostic value are raised by the multitude of pathological conditions associated with an elevated level of this protein. When determining its elevated level, the geriatricians cannot state whether the cause is diabetes, heart failure, pancreatitis, cancer, or depression. It seems that this parameter is not specific, thus is of little diagnostic value.

The second of the analyzed receptors for AGE, i.e., receptor for advanced glycation end products (RAGE), is the protein responsible for inflammation, apoptosis, reactive oxygen species (ROS) signaling, proliferation, and autophagy [20]. sRAGE is a soluble form of protein that circulates in blood plasma competing for ligands, thus having RAGE-reversing, anti-inflammatory properties [18,48]. We expected a correlation between sRAGE and pentosidine, glucose, or HbA1c. These expectations have not been met, but it turned out that sRAGE correlates with triglycerides and SR-BI in our study population. Both of these parameters are related to lipid metabolism—TAG obviously, SR-BI due to the fact that, in addition to AGEs, it also binds HDL, mediating selective uptake of cholesterol esters and HDL-dependent cholesterol efflux [19]. So far, no one has associated SR-BI and sRAGE. Only Marshe et al. found that sRAGE is related to another scavenger receptor CD36 and blocks CD36-mediated uptake of LDL modified by hypochlorous acid and in this way reduces foam cell formation [48]. Thus, it appears that both sRAGE and SR-BI have an inhibitory effect on the development of atherosclerosis, and it is beneficial for the patient if the levels of these proteins, due to their mutual correlation, are high.

In our study group, the negative correlation between SR-BI and HDL was also observed. This is an expected result, as it has been proven that elevating SR-BI expression causes an increase in HDL clearance, a reduction in plasma HDL levels, and an increase in biliary cholesterol levels [19].

The analysis revealed also that SR-BI correlates with one more parameter—Lectin-type oxidized LDL receptor 1. In the latest publications (2020 and 2021), LOX-1 is presented as a promising target for early diagnosis and cardiovascular risk prediction [49,50,51]. Such enthusiastic announcements made us expect to find higher levels of LOX-1 in people with stroke, after a myocardial infarction or with obesity, but no such association was found in our geriatric population, perhaps due to a small enough study group or a too-high degree of atherosclerosis in the entire study group. Nevertheless, the correlation found between LOX-1 and SR-BI is a novelty in the subject literature, and it is worth continuing research on the cause-and-effect relationship between these two receptors, since they correlate with each other.

Through modeling, we found that for SR-BI, HDL is a good predictor. West reported this relationship for subjects with hyperalphalipoproteinemia [52]. This is the first time that this dependence is described for the geriatric population. However, this finding is not clinically valuable, because it is still more convenient to measure HDL levels in patients’ blood serum.

Metformin is the most prescribed glucose-lowering medicine worldwide and has been used to treat diabetes since 1957 [53]. In the late 20th century, it was proposed that metformin may act as a glycation inhibitor [54]. Some research results have suggested that the drug binds to the α-dicarbonyls, methylglyoxal (MG) or 3-deoxyglucosone, preventing these components from taking part in the course of glycation [55]. This theory is currently being challenged by concluding that AGE-inhibiting effects result from an improvement in glycemic control rather than from a specific dicarbonyl detoxification [56]. The results of this project did not show that pentosidine is reduced in patients who take metformin, which may be an indication that metformin does not actually inhibit glycation, but nevertheless, more targeted studies should be performed to conclusively confirm this statement.

Numerous data from in vitro experiments indicate that metformin inhibits the expression of LOX-1. This is evidenced, for example, by the data of Hung et al., who studied this dependence in human umbilical vein endothelial cells (HUVECs) [57]; by Shiu et al., who examined the relationship on human aortic endothelial cells incubated with AGE-BSA [58]; and by Ouslimani et al., who investigated bovine aortic endothelial cells [59]. The above-mentioned studies showed that metformin protects against oxLDL-induced endothelial apoptosis, oxLDL-induced intracellular calcium rise, and mitochondrial dysfunction [57], as well as reduces, in a dose-dependent manner, the expression of LOX-1 both in stimulated (by either glucose or AGE) and in unstimulated cells [59]. No study of this relationship has been found in patients’ trials; thus we are probably the first to report it. Our research shows, however, that treatment with metformin alone does not cause a large difference in the concentration of LOX-1 in the serum; similarly with insulin, only the combination of metformin with insulin seems to cause patients to demonstrate a significantly reduced LOX-1 serum level. This is a very important finding that can help geriatricians make therapeutic decisions.

Free amine groups are potentially available to react with a reducing sugar; therefore, this is an important parameter regarding the potential for glycation. In addition, Valencia et al. found that free amine content measured using OPA correlates strongly with RAGE binding affinity to AGE [60]. Fracesso expected differences between the amount of free amine content in the serum and tissues of rats after myocardial infarction compared to controls but did not find them [26]. It seems that this is the first time this has been observed in geriatric patients. Higher levels of serum free amino groups were found in the study patients with ischemic heart disease and after stroke. These are diseases associated with atherosclerosis. We expect that if there were more people in the test group with history of myocardial infarction, we would also see a difference with the control group. Therefore, it can be concluded that patients with cardiovascular diseases have an increased level of free amine content in the serum, and thus a greater potential for glycation and probably greater binding affinity RAGE to AGE. To increase the credibility of these statements, the study should be repeated on a larger geriatric population.

In addition to free amine content, we also observe higher pentosidine values in patients with ischemic heart disease. This was an expected result, as it was previously reported that circulating fluorescent AGE and pentosidine levels have been associated with cardiovascular disease [61,62,63,64,65]. The same applies to the relationship of pentosidine with arterial hypertension. These dependencies are so strong that IHD and hypertension can be considered as a predictor of high pentosidine levels. These correlations have been described earlier [64,66,67]. This is a logical consequence of the fact that glycation causes the formation of cross-links between collagen fibers and other proteins responsible for biochemical properties in the arteries, making them more rigid, unable to dilate properly, and manifesting in hypertension [68].

It is stated that the kidney normally clears circulating AGEs, but these products accumulate both in diabetic and not-diabetic nephropathy [69]. Pentosidine correlates positively with creatinine and negatively with GFR in our study. A correlation similar to the one we found was reported by Dozio et al. based on patients with chronic kidney disease [69], Yoshida based on patients who underwent renal transplantation [61], and Schalkwijk comparing hemodialyzed patients with normal subjects [70]. We would like to point out that there were generally no subjects with renal failure in our study group, so one may wonder whether pentosidine is an indicator showing subclinical renal changes.

We do not try to present fluorescence of pentosidine as a diagnostic tool, but we want to pay attention to the low cost, speed, and simplicity of AGEs determination by measuring fluorescence compared to other methods, such as enzyme immunoassay or liquid chromatography with tandem mass spectrometry (LC-MS/MS). For research purposes, this method seems to be sufficient.

The glycation pathway is associated with reactive oxidative species (ROS) and with peroxidation. ROS peroxidize membrane unsaturated fatty acids, leading to the generation of reactive aldehydes as advanced lipid peroxidation end products (ALE). Resulting reactive aldehydes react with proteins to cause an alteration of protein structure to exacerbate the complication of diseases such as diabetes and atherosclerosis [71]. One of the classic products of lipid peroxidation is malondialdehyde, which is easily determined in serum and tissues [72]. For many years, laboratory animals and human studies have analyzed the effect of hypoglycemic therapies on oxidative stress and the intensity of lipid peroxidation. Depending on the research model, the results vary. For example, Sotoudeh showed in a study with diabetic rats that metformin reduces MDA levels [73]; Kocer claimed the same while looking at the results of women with polycystic ovaries [72]; yet Alrefai reported that metformin does not lower MDA in obese patients with T2DM [74]. Similarly, in the case of insulin, there have been reports of both an anti-inflammatory role [75] as well as that hyperinsulinemia seems to cause an exaggeration of oxidative stress [76]. Bunck et al. confirmed that insulin therapy alone for one year does not cause a decrease in MDA levels [77]. Many researchers recommend a combination therapy with insulin as the best therapeutic approach for reducing oxidative stress in patients who need glycemic control [74,76,78]. The results of our study on the group of geriatric patients confirm that treatment with both metformin and insulin causes a decrease in the concentration of the lipid peroxidation marker.

## 5. Conclusions

The diagnosis of diseases of old age is constantly developing. Today, we have the ability to use parameters that were unavailable even several or several dozen years ago, because basic sciences discovered their importance only relatively recently. The mentioned glycated hemoglobin came into use at the end of the 20th century [79], and nowadays galectin 3 is slowly being introduced into diagnostics [33,35]. Research such as this is also helpful in framing therapeutic strategies. The importance of glycation in the development of degenerative diseases is becoming more and more obvious; therefore, attempts have been made for a long time to find an appropriate glycation inhibitor [54,55,80]. Recently it has been even postulated that galectin 3 inhibitors may be helpful in the treatment of COVID-19 [81].

In this project, it was shown that glutathione and SR-A have no noticeable value as indicators of typical geriatric diseases, but other parameters appear to be interestingly related to pathologies of old age. According to the study, pentosidine is significantly increased in ischemic heart disease and arterial hypertension and is also associated with kidney function. We also found indications that sRAGE and SR-BI have anti-atherogenic effects. The role of galectin 3 and LOX-1 in the pathomechanism of senile diseases is still under investigation. In our study, we observed a difference in the concentration of these proteins depending on the treatment. Combined treatment with insulin and metformin seems to be particularly beneficial as it reduces the intensity of lipid peroxidation and probably influences the expression of the protein responsible for lipoprotein binding, thus having the potential to inhibit atherosclerosis. Metformin alone results in a decrease in galectin 3, which may protect against fibrosis and organ failure, such as the kidneys and the heart.

As emphasized several times in the discussion, this research should be repeated on a larger geriatric population. However, this is likely a great value in this research direction due to the possibility of finding a parameter that allowed for an even earlier diagnosis of prediabetic conditions or other numerous diseases of old age, as well as the possibility of monitoring the treatment of metabolic diseases typical of the elderly.

## Figures and Tables

**Figure 1 ijerph-19-07524-f001:**
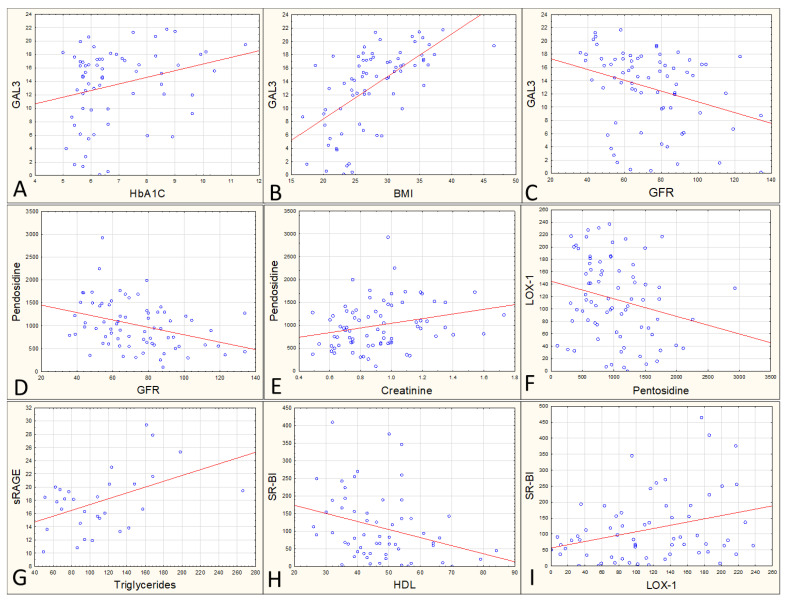
Scatterplots for selected pairs of analyzed parameters. Scattering of results between GAL3 and HbA1c (panel **A**), between GAL3 and BMI (panel **B**), between GAL3 and GFR (panel **C**), between pentosidine and GFR (panel **D**), between pentosidine and creatinine (panel **E**), between pentosidine and LOX-1 (panel **F**), between sRAGE and triglycerides (panel **G**), between SR-BI and HDL (panel **H**), and between LOX-1 and SR-BI (panel **I**).

**Figure 2 ijerph-19-07524-f002:**
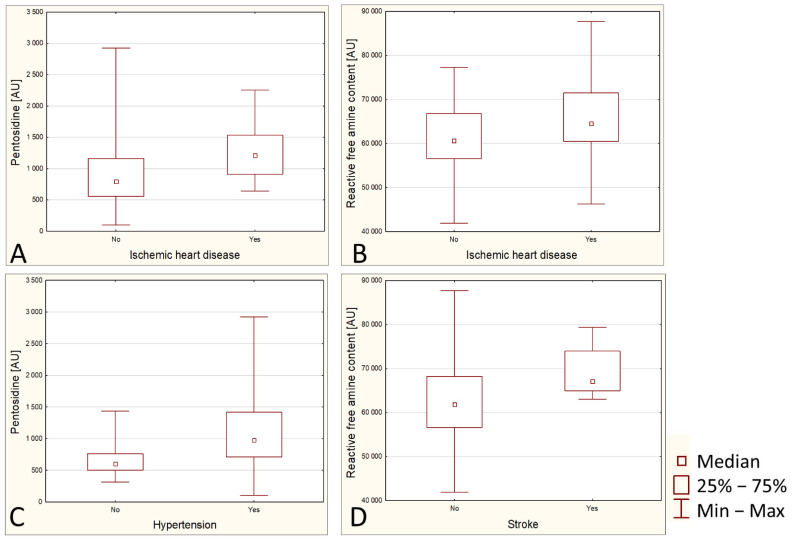
Group comparison between variables (box and whisker plots). Comparison of pentosidine (panel **A**) and reactive free amine content (panel **B**) in patients with and without ischemic heart disease. Comparison of pentosidine (panel **C**) in patients with and without hypertension. Comparing the level of reactive free amine content in patients with and without stroke (panel **D**).

**Figure 3 ijerph-19-07524-f003:**
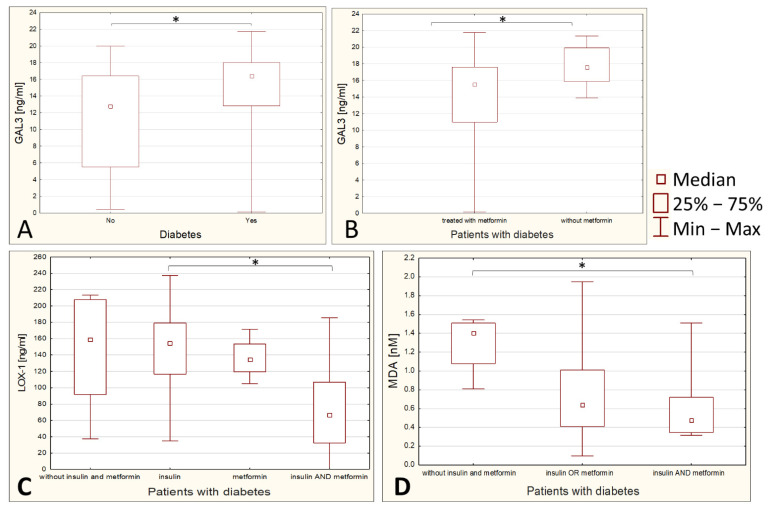
Group comparison between variables (box and whisker plots). Comparison of GAL3 levels in patients with and without diabetes (panel **A**). Comparison of GAL3 levels in patients taking and not taking metformin (panel **B**). Comparison of LOX-1 levels in diabetic patients not treated with insulin or metformin, those treated with insulin, those treated with metformin, and those treated with both medications simultaneously (panel **C**). Comparison of MDA levels between diabetic patients not treated with insulin or metformin, those treated with insulin or metformin, and those treated with both medications simultaneously (panel **D**). An asterisk (*) indicates between which groups the difference is statistically significant.

**Table 1 ijerph-19-07524-t001:** Characteristics of the study group.

	Without Diabetes 0 (N = 38)	With Diabetes 1 (N = 41)	*p*
Age (Max-Min)	79.65 (64.0–92.0)	78.7 (64.0–94.0)	0.52
Sex (Women/Men)	29/9	30/11	0.7480
BMI (Max-Min)	26.21 (17.4–46.6)	29.43 (18.7–38.6)	0.003
Ischemic heart disease 0/1 *	27/9	25/12	0.4831
Myocardial infarction 0/1 *	37/1	35/6	0.0667
Stroke 0/1 *	32/5	36/5	0.5634
Hypertension 0/1 *	11/27	6/35	0.1014
Peripheral artery disease 0/1 *	34/3	37/4	0.5582
Atherosclerosis 0/1 *	13/25	9/32	0.2245
Hyperlipidemia 0/1 *	18/19	28/12	0.0563
Fatty liver disease 0/1 *	32/6	30/11	0.2329
HbA1c (%)	5.81 ± 0.39	7.69 ± 1.54	<0.001
Glucose (mg/dL)	98.39 ± 17.35	155.85 ± 71.59	<0.001
Hemoglobin (g/dL)	12.64 ± 1.54	12.72 ± 1.31	0.8365
Cholesterol (mg/dL)	202.08 ± 54.04	151.18 ± 41.88	<0.001
HDL (mg/dL)	53.69 ± 10.75	41.05 ± 1.68	<0.001
LDL (mg/dL)	124.94 ± 44.57	81.54 ± 32.68	<0.001
Triglycerides (mg/dL)	114.32 ± 47.84	135.42 ± 57.45	0.0971
Total protein (g/dL)	6.98 ± 0.57	7.18 ± 0.78	0.2056
CRP (mg/L)	3.44 ± 9.02	4.37 ± 5.91	0.5916
Creatinine (mg/dL)	0.90 ± 0.22	0.93 ± 0.28	0.5572
Uric acid (mg/dL)	5.58 ± 1.38	6.29 ± 1.99	0.0975
GFR (mL/min/1.73 m^2^)	72.95 ± 20.34	72.44 ± 22.67	0.9210

* Number of patients without a feature/number of patients with a given feature.

**Table 2 ijerph-19-07524-t002:** Mean contents and standard deviation of the analyzed parameters in the serum of the studied geriatric patients.

Parameter (N)	Mean	SD
Pentosidine (AU) (81)	1015.26	513.17
Free amine content (AU) (82)	62,838.79	8247.55
GAL3 (ng/mL) (82)	13.23	5.77
LOX1 (ng/mL) (78)	115.47	64.51
sRAGE (pg/mL) (32)	17.98	4.74
SR-BI (SCRAB1) (pg/mL) (63)	116.96	103.34
SR-A (MSR1) (ng/mL) (75)	2.58	4.04
MDA (nM) (78)	0.75	0.51
Glutathione (mM) (81)	0.15	0.10

## Data Availability

Raw data are available from the corresponding author in response to a written request.

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
