# Peer review of "How Diabetes and Other Comorbidities of Elderly Patients and Their Treatment Influence Levels of Glycation Products"

_ijerph, 2022, doi:10.3390/ijerph19127524_

Round 1

Reviewer 1 Report

1. Introduction

“Gaining detailed knowledge about the factors leading to the aging process and the course
of diseases occurring in the elderly (including diabetes) makes it possible to
develop a way to delay the decline in the functionality of the human body and
at the same time extend life.”

This statement / intention might be too ambitious. The authors should clearly define their
aims/goals of this specific study.

2. Materials and Methods

“The research material were serum samples taken during hospitalization at the Department of
Geriatrics, University Teaching Hospital in WrocÅ‚aw.”

Please indicate why these patients were admitted to the hospital. Please also indicate
detailed in- and exclusion criteria of this specific study. How was patient
selection done?

Table 1: In the opinion of the reviewer, the formatting of the table is confusing (e.g.
K/M, 0/1). The specific statistical test that has been used must not appear directly
in the table. Please clarify how the diagnosis “atherosclerosis” is defined.

3.Results

Main conceptual axis: In the opinion of the reviewer, the main conceptual axis of
the results is not clear. The authors should specify / clarify their intentions.
While going through the results section, it does not become clear why some analytes
are presented, while others are not.
In my opinion the results that have been made should be presented from a clinical point
of view, thereby focusing on different treatment regimes in geriatric patients
with diabetes (a cautious attempt of this has been tried in Figure 3).

Table 3: In the opinion of the reviewer, the layout of table 3 is confusing as well.
I would suggest using a heat map instead and provide all these numbers as a
supplementary table.

199: wrong spelling “Glutatnion“

Reviewer 2 Report

-          The title is recommended to be more specific.

-          The design of the study is missing.

-          The enrolment of the participants should be explained (i.e. whether participants were randomized or they were included in a consecutive manner).

-          Inclusion and exclusion criteria are missing.

-          Were there any smokers among participants?

-          Did participants use any other medications (antihypertensives, antilipemics,...)? Which ones? This should be clearly stated.

-          When the blood samples were drawn (i.e. in the morning, after an overnight fasting)? At least 8 h of fasting is recommended for the determination of biochemistry analyses.

-          How GFR was obtained? This should be explained in the Methods section. If it is estimated by formula (e.g. MDRD, CKE-EPI...?) which one is used and the reference should be provided.

-          Sample size and power of the study need to be addressed.

-          The data on HbA1c, glucose, hemoglobin, lipid parameters, total protein, CRP, cre, uric acid, GFR are missing in Table 1.

-          Lines 191, 199 and in Table 1: there are some typographical errors (i.e. (p=0.267, p=0.031), „Glutatnion“, „Sex (K/M)“).

-          Lines 308-309: the Authors should re-written this part of the text more clearly, taking care of correct quotations: „It would seem that galectin 3 is a very interesting diagnostic tool, and many authors present it as a novel biomarker, np. Alves, Hrynchyshyn, Kanukurti, King [31,32,34,37].“

-          Instead of “diabetics” in Figure 3, the Authors should better write “patients with diabetes”.

Round 2

Reviewer 1 Report

Table 1: Please add (if available) information about heart failure diagnosis (HFpEF, HFrEF)

Discussion: In your study, metformin treatment was associated with lower systemic galectin-3 (already shown in DOI: 10.1210/jc.2009-1619 , please add). Galectin-3 is an interesting biomarker and target in CVD. Since metformin reduces cardiovascular mortality, all-cause mortality and CV events in CAD patients the authors might consider recent studies that tried to link a lower thrombogenicity with metformin treatment in their discussion: DOI: 10.1093/eurheartj/ehac128 and DOI: 10.1093/eurheartj/ehac034 and DOI: 10.1007/s10557-020-07040-7.

Author Response

Thank you for the positive evaluation of our work. We appreciate all comments and tips. Responding to comments from the second round of review:

Note 1: „Table 1: Please add (if available) information about heart failure diagnosis (HFpEF, HFrEF)”

Response 1: Heart failure with preserved ejection fraction (HFpEF) was diagnosed in some of the study participants. However, cardiac ultrasound was not a routine examination, it was performed only in patients with clinical evidence. For this reason, data on the type of heart failure were not analyzed.

Note 2: „Discussion: In your study, metformin treatment was associated with lower systemic galectin-3 (already shown in DOI: 10.1210/jc.2009-1619 , please add). Galectin-3 is an interesting biomarker and target in CVD. Since metformin reduces cardiovascular mortality, all-cause mortality and CV events in CAD patients the authors might consider recent studies that tried to link a lower thrombogenicity with metformin treatment in their discussion: DOI: 10.1093/eurheartj/ehac128 and DOI: 10.1093/eurheartj/ehac034 and DOI: 10.1007/s10557-020-07040-7.”

Response 2: Thank you for this suggestion. We added this thread in the discussion.

Thank you again for your time and the opportunity to improve the manuscript.

Yours faithfully

Authors

Reviewer 2 Report

The Authors have mostly made corrections according to the Reviewer's suggestions.

The units for some parameters in Table 1 are missing.

Author Response

Thank you for the positive evaluation of our work. We appreciate all comments and tips. Responding to comments from the second round of review:

Note 1:The units for some parameters in Table 1 are missing.

Response 1: We apologize for this oversight. We have already completed this data.

Thank you again for your time and the opportunity to improve the manuscript.

Yours faithfully

Authors